# Multiplicative joint coding in preparatory activity for reaching sequence in macaque motor cortex

Tianwei Wang [1,2,3], Yun Chen [1,2,3], Yiheng Zhang [1,2,3] & He Cui [1,2,3,4] ✉

Although the motor cortex has been found to be modulated by sensory or cognitive sequences, the linkage between multiple movement elements and sequence-related responses is not yet understood. Here, we recorded neuronal activity from the motor cortex with implanted micro-electrode arrays and single electrodes while monkeys performed a double-reach task that was instructed by simultaneously presented memorized cues. We found that there existed a substantial multiplicative component jointly tuned to impending and subsequent reaches during preparation, then the coding mechanism transferred to an additive manner during execution. This multiplicative joint coding, which also spontaneously emerged in recurrent neural networks trained for double reach, enriches neural patterns for sequential movement, and might explain the linear readout of elemental movements.

The motor cortex has long been thought to be central in planning and generating movement. A large body of evidence demonstrates a correlation between neuronal activity in the motor cortex and a variety of motor variables, such as direction, speed, distance, and trajectory[1–7]. Beyond the single ballistic movements examined in these studies, multi-step movements, such as sequencing and ordering action, are crucial in daily behavior[8,9]. As one of the brain areas conveying highly accurate information about movement timing[10] and kinematics[11], the motor cortex seems to be involved in causal sequencing of multi-step movements[12]. Sequential information has been reported to be encoded in the population response before movement initiation[13–15]. In addition, most neurons are reported to show activity related to both target location and serial order[16,17]. However, most of these studies instructed the sequence of movement with serial sensory stimuli, which might result in neural activity that differs from internally generated motor sequences[18–20]. In tasks carried out in the absence of serial sensory inputs, neuronal activity related to sequential contexts emerges during preparation, and becomes prominent during execution[21,22]. Furthermore, despite differences at the single-neuron level, the neural population preserves a reliable readout of movement direction. That is to say, both individual movement elements and sequential information are simultaneously and robustly encoded in the motor cortex[21].

In principle, a continuous action sequence consists of elements spatio-temporally coordinated in a complex manner, rather than a series of independent actions[23–25]. However, the "competitive queuing" hypothesis suggests that the brain produces sequential movement via a combination of parallel coding of specific actions[26]. A recent study on double reach supports this parallel coding hypothesis, suggesting that the motor cortex does not fuse two reaches, but recruits two independent motor processes sequentially[27]. The resulting concurrence of motor execution and motor planning, however, is insufficient for rejecting the possibility of interaction between movement elements beforehand. It remains unclear if sequential movement is parallel or jointly coded in the preparation period.

To further explore the motor preparation and encoding characteristics of sequential movements in a strict behavioral and neurophysiological context, we recorded neuronal activity from the motor cortex via implanted arrays or single electrodes while monkeys were performing a double reach that was instructed by simultaneously presented cues that had to be memorized. We found that neuronal activity could be regressed as a multiplication of directional tunings to reaching

[1]Institute of Neuroscience, Key Laboratory of Primate Neurobiology, Center for Excellence in Brain Science and Intelligence Technology, Chinese Academy of Sciences, Shanghai 200031, China. [2]University of Chinese Academy of Sciences, Beijing 100049, China. [3]Chinese Institute for Brain Research, Beijing 102206, China. [4]Shanghai Center for Brain Science and Brain-inspired Technology, Shanghai 200031, China. ✉e-mail: hecui@cibr.ac.cn

elements in the preparatory period, and then converted to parallel coding for both movement elements after movement onset, indicating the existence of a gain-like interaction in planning the motor sequence. Neural population dynamics derived from our array-recorded data indicates that a nonlinear interaction is embodied in the spatial structure of initial states. In computational simulations, multiplicative coding for motor sequences spontaneously emerges in a recurrent neural network, and benefits reliable linear readouts of movement elements. These results suggest that the motor cortex is profoundly involved in concatenating multiple movement elements into a sequence, and that a gain-like multiplication is a key signature of complex serial behavior.

## Results
### Behavioral task

Three rhesus monkeys (*Macaca mulatta*, male 5–10 kg) performed the memory-guided double-reach task (Fig. 1a). A trial began with a green dot displayed on the center of a touch screen, and the monkey was required to touch it. After 300 ms, in 1/3 of the trials (single-reach, SR), another green dot was presented as a reaching goal for 400 ms (cue period) at one of the six corners of a regular hexagon (i.e., at directions of 0°, 60°, 120°, 180°, 240°, or 300°). After the peripheral cue was extinguished, there was a memory period of 400–800 ms. Thus, the

total delay from Cue to GO was 800–1200 ms. The monkey was trained to keep its hand on the central green dot until it was turned off (GO signal), and then reach the previously cued location to obtain a reward. In the remaining trials (double-reach, DR), a green square and a green triangle were presented simultaneously during the cue period. The square was in the same alternative directions as the SR surrounding targets. The triangle was displaced from the square by 120° clockwise (CW, 1/3 of trials) or 120° counterclockwise (CCW, 1/3 of trials). After the memory period without peripheral cues, the monkey was required first to reach the memorized square location, and then to immediately reach the memorized triangle location. The monkey was rewarded only if it reached the specified target within a margin of three centimeters, and in the correct order. For a correct trial, the green square would reappear after the first reach, and the triangle would appear in purple after the second reach. All 18 conditions (three trial types × six directions) were pseudo-randomly interleaved. Only correct trials were included in the analysis. Event markers are denoted as the GO signal (GO), the first/only movement onset (MO), the first/only movement end (ME), and the second movement onset (MO2).

Hand trajectories exhibited a stereotype movement pattern in each condition for well-trained monkeys. All first reaches started from

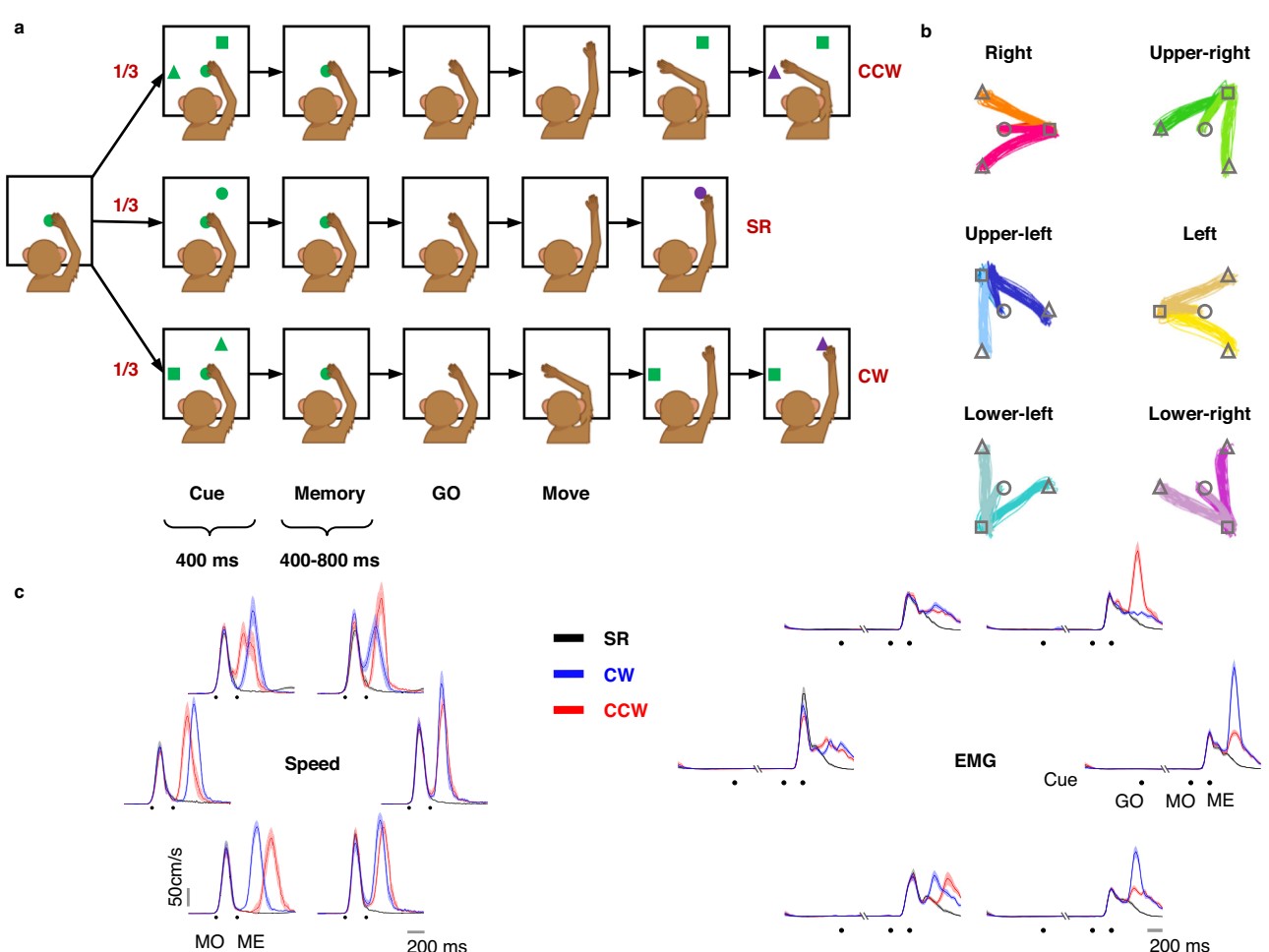

**Fig. 1 | Paradigm and behavior. a** Three types of trials were pseudo-randomly interleaved in each session. In single-reach (SR) trials, monkeys had to perform memory-guided center-out reach. In double-reach (DR) trials, two targets (a square and a triangle) were presented simultaneously in the cue period, and then extinguished; the monkeys were required to hold the central target for a 400–800 ms memory period until it was turned off (GO signal). Next, monkeys finished reaching both targets in the sequence of the square to the triangle within 700–1200 ms. The triangles were located 120° from the squares in CW or CCW directions. Monkey

cartoons were created by Miss Jiayue Li. **b** Hand trajectories in different conditions are grouped by their 1st/only reach direction from monkey C. Some trajectories are overlapped due to high similarity. No significant difference was found before the end of 1st/only reach (one-way ANOVA, $p > 0.05$). **c** Surface electromyography (sEMG) and speed in one typical session. The Pearson correlation coefficient of the speed profile until the first movement end between double reach and single reach was $0.99 \pm 0.006$ (mean ± sd), and of sEMG of extensor digitorum communis (EDC) was $0.99 \pm 0.005$ (mean ± sd) for monkey C.

the center and moved towards the corresponding target in each condition (Fig. 1b). Muscular activities remained constant during the preparatory period across different conditions, excluding the possibility that the monkeys might develop different premature movements (e.g., adjust arm orientation) after cue for different conditions. The Pearson correlation coefficient of speed profiles until ME between DR and SR was $0.99 \pm 0.006$ (mean ± sd), and of surface electromyography (sEMG) of extensor digitorum communis (EDC) was $0.99 \pm 0.005$ (mean ± sd) for monkey C (Fig. 1c). In addition, the dwell time on the first target was $194 \pm 75$ ms (mean ± sd) for monkey C, $350 \pm 110$ ms (mean ± sd) for monkey G, and $150 \pm 47$ ms (mean ± sd) for monkey B. The duration of DR was $586 \pm 95$ ms (mean ± sd) for monkey C, $818 \pm 131$ ms (mean ± sd) for monkey G, and $481 \pm 72$ ms (mean ± sd) for monkey B, averaged across conditions. These results verified the expected transitory dwell on the first target in this task, and indicated behavioral consistency between SR and the first reach of DR in the same direction, in terms of hand trajectory, speed profile, and sEMG.

## Heterogeneity in neuronal activity indicated mixed selectivity

All electrophysiological recording sites were in the hemisphere contralateral to the hand used during the task. Only one hand was used by monkeys B and G, but for monkey C data were recorded first with single electrodes, and then arrays in the other hemisphere with a switch of hands. We collected 322 well-isolated task-related neurons from single-electrode recordings (224 from monkey B, 98 from monkey C left hemisphere) and 162 units sorted from array recordings (44 from monkey G, 118 from monkey C right hemisphere) in the motor cortex (Supplementary Fig. 1). Among these, we found considerable heterogeneity in firing patterns. Figure 2 illustrates four representative cells. The neuron in Fig. 2a exhibited a two-peak firing pattern in DR, each peak after movement onset, while it had only one burst in SR. Notably, the direction with the highest firing rate changed remarkably in sequential movements. The neuron in Fig. 2b fired with a constant preferred direction (PD) towards the lower left. Surprisingly, even though its directional selectivity was remarkably similar for both SR and DR, the firing rate was significantly higher in DR (according to the 95% confidential interval plotted in the shade), indicating that it conveyed information regarding target-movement number. Also, the preparatory activity would diverge with the 2nd reach before GO and MO in neurons, as in Fig. 2c, d.

We further examined the proportion of neurons with sequence selectivity in three periods: preparatory (600 ms before GO), pre-movement (200 ms before MO), and peri-movement period (200 ms before ME). Among the 322 neurons recorded by single-electrodes, 52% exhibited significantly different firing rates for SR and DR in the preparatory period (Wilcoxon rank-sum test, $p < 0.05$). This proportion increased to 68% in the pre-movement period, and then to 84% in the peri-movement period (Wilcoxon rank-sum test, $p < 0.05$). As for the comparison between CW and CCW trials, 30%, 48%, and 72% of neurons showed significant differences during the preparatory, pre-movement, and peri-movement periods, respectively (Wilcoxon rank-sum test, $p < 0.05$). For the 162 array-recorded neurons, 76%, 86%, and 95% were significantly tuned to sequence during preparatory, pre-movement, and peri-movement periods, respectively (Wilcoxon rank-sum test, $p < 0.05$). In comparing CW and CCW trials, the proportions were 46%, 66%, and 86% during the preparatory, pre-movement, and peri-movement periods, respectively (Wilcoxon rank-sum test, $p < 0.05$). These considerable proportions reveal a substantial sequence selectivity in the motor cortex.

## Additive vs. multiplicative joint coding

The above results show single-neuron responses related to reaching sequences. However, whether such sequence-related responses result from joint coding or parallel coding is the next question. Then, based on the directional tuning function:

$$FR = a\cos(\theta - \theta_{PD}) + c \qquad (1)$$

where $\theta$ is the movement direction, $\theta_{PD}$ is the PD, $a$ and $c$ denote regression coefficients; we developed two fitting models.

The parallel coding assumes the sequence-related difference comes from the overlap of two independent tuning components. In this model, sequential modulation is a parallel process resulting from the preparation of the second movement while the first movement still is in flight, as pointed out by ref. 28 Here, we focused on directional tuning alone, and defined an 'additive model' as follows:

$$FR = a_1\cos(\theta_1 - \theta_{PD}) + a_2\cos(\theta_{21} - \theta_{PD}) + c \qquad (2)$$

where FR is neuronal firing rate, $\theta_1$ is the movement direction of the first reach, $\theta_{21}$ is the second movement direction starting from the first reaching endpoint, that is, in execution coordinates (Fig. 3a), since the regression result (Fig. 3b) indicates that the second reach is predominately conveyed in execution coordinates (movement direction, $\theta_{21}$ in Fig. 3a) rather than visual coordinates (target location, $\theta_2$ in Fig. 3a). $\theta_{PD}$ represents the intrinsic PD, $a_1$ and $a_2$ are coefficients, and $c$ is the baseline firing rate. For simplicity, we assumed the PD to be consistent for both terms at the same time.

However, since the visual targets in our task were presented simultaneously, rather than sequentially as in many previous studies[9,15,16,28], the monkeys were more likely to prepare the entire reaching sequence beforehand[19,24]. In this case, the different responses in DR might not simply result from the overlap of the "preparation-execution", but from interaction between the tuning components corresponding to two reaches. Therefore, this raises the possibility of joint coding, for which an interactive term is essential. For computational convenience, and as inspired by a previous study suggesting that hand speed may act as a "gain field" to the directional cosine tuning function[29], we propose a "multiplicative model" to depict the potential nonlinear gain-modulation between both elemental movements:

$$FR = a_1\cos(\theta_1 - \theta_{PD}) + b\cos(\theta_{21} - \theta_{PD})\cos(\theta_1 - \theta_{PD}) + c \qquad (3)$$

where $b$ is a coefficient and other notations as in Eq. 2. If we set $\Delta\theta = (\theta_{21} - \theta_1)/2$, then the multiplicative term in Eq.3 can be transformed into a summation form that includes a doubled frequency (Eq. 4).

$$b\cos(\theta_1 - \theta_{PD})\cos(\theta_{21} - \theta_{PD}) = \frac{b}{2}\cos\left(2\left(\theta_1 - \theta_{PD} + \frac{\Delta\theta}{2}\right)\right) + \frac{b}{2}\cos\Delta\theta \qquad (4)$$

To further examine the interaction between element movements and to avoid overfitting in the regression analysis, in addition to the standard paradigm described in Results (Fig. 1a), we trained monkey C to perform an extended version of the task with multi-angle, in which the angle between the square and triangle could be 60° or 120° in both CW and CCW directions as well as 180°. This multi-angle task has 36 conditions in total (six SR and 30 DR).

We tested these two possibilities on condition-averaged normalized firing rates with a 200-ms sliding window[30]. The fitting results of an example neuron are shown in Fig. 3c, in comparison with its actual PSTHs. This neuron obviously had a sequence-related mixed selectivity, because its peri-movement activity varied with different subsequent movements, and the preparatory activity was also condition-dependent, though with small variation. The response reconstructed by the additive model (Eq. 2) reproduced the peri-movement firing pattern, but it did not capture the sequence-specific modulation during preparation. In contrast, the multiplicative model (Eq. 3) better

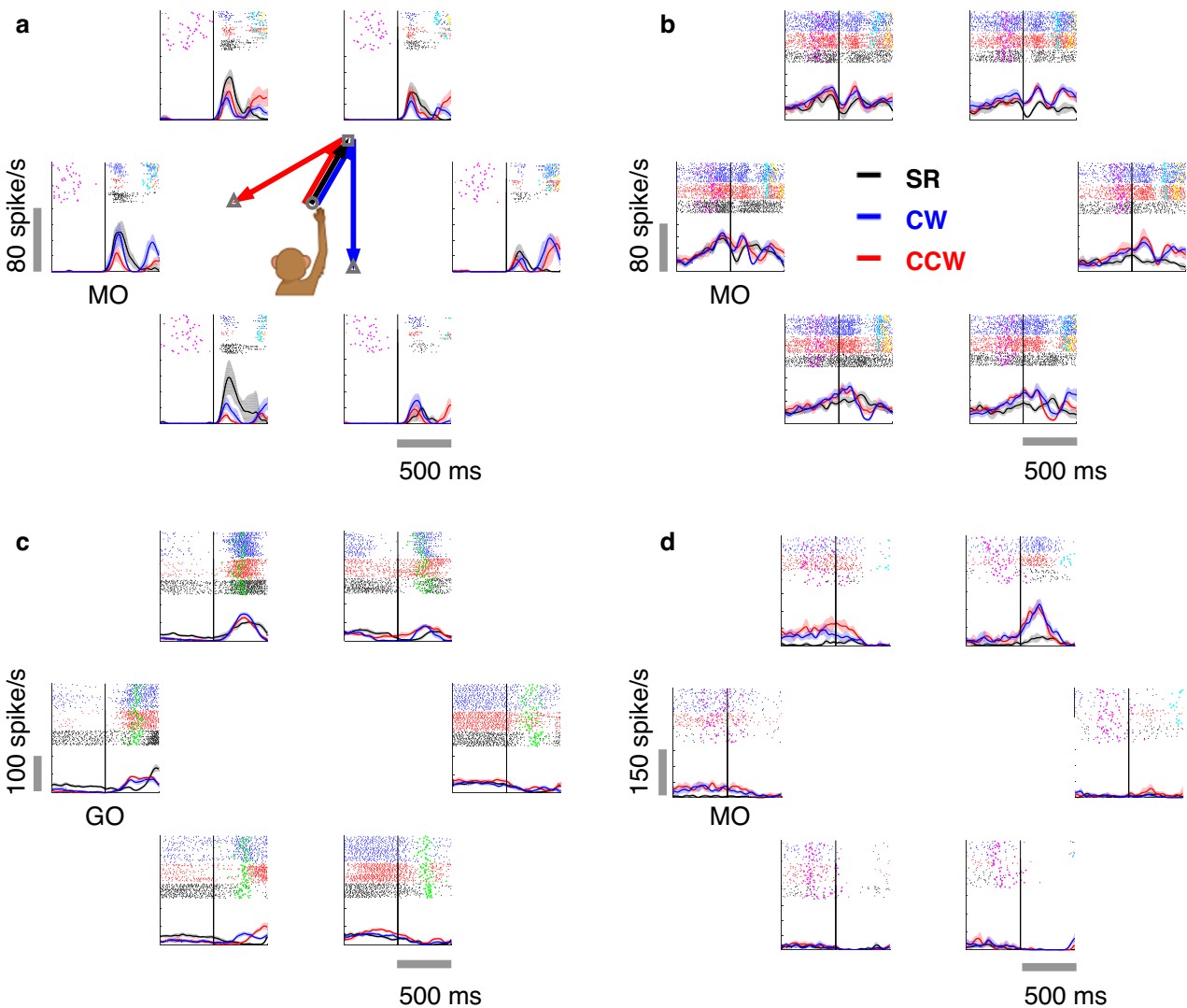

**Fig. 2 | Examples of cells in the motor cortex showing heterogeneous firing patterns.** In each panel (**a**–**d**), the six subplots show PSTHs of the same neuron in three conditions with the first reach toward the corresponding location (e.g., the upper-right subplot denotes the first reach to 60°). Rasters are plotted at the top of each PSTH (20-ms SD Gaussian kernel, mean ± 2 standard error). Spike trains in SR (black line), CW (blue line), and CCW (red line) trials are aligned to the first/only movement onset (MO) in (**a**, **b**, **d**), but aligned to GO-cue in (**c**). The time of GO (magenta dots), MO (green dots), the second movement onset (MO2, cyan dots), and the second movement end (yellow dots) are presented in the raster. Monkey cartoons were created by Miss Jiayue Li.

captured neural activity during the preparatory period, while losing that during the peri-movement period. In Fig. 4, we plotted directional tuning curves of the same example cell with its actual firing rates (Fig. 4, left panel), along with reconstructed firing rates by additive (Fig. 4, middle panel) or multiplicative (Fig. 4, right panel) models. The real firing rate for plotting and fitting was normalized and averaged around MO (−100–100 ms to MO, peri-MO) and around ME (100–300 ms to MO, peri-ME), respectively. Here, we assume the neuron has the same $\theta_{PD}$ in a certain time bin across conditions in each model. However, because of the modulation of 2nd reach directions (whether multiplicative or additive), the apparent PD (i.e., the direction with the highest FR) may change compared to the $\theta_{PD}$, for peri-MO (Fig. 4a), the neural tuning curves consisted mostly of two peaks and were only replicated by the tuning curves of the multiplicative model. This was not accidental, because frequency doubling is a corollary of the product of two trigonometric functions (Eq. 4). For peri-ME (Fig. 4b), apparent PD shifted with conditions in data, and only the additive model yielded a similar outcome. These results suggest that different coding rules cause distinctly different firing patterns. The multiplicative interaction contributes to the period changing, whereas

the additive relation can easily lead to PD shifts while retaining the periodic identity. Comparing two epochs, the two coding possibilities could co-exist and might alternate.

To further investigate the temporal dynamics of joint-coding rules, we proposed a "full model" to combine the two modulation forms:

$$FR = a_1 \cos(\theta_1 - \theta_{PD}) + a_2 \cos(\theta_{21} - \theta_{PD}) + b \cos(\theta_{21} - \theta_{PD}) \cos(\theta_1 - \theta_{PD}) + c \tag{5}$$

where descriptions of notations are the same as in Eq. 2 and Eq. 3, defining $a_1$ as the first reach weight, $a_2$ as the additive weight, and $b$ as the multiplicative weight. The fluctuation of the regression coefficients ($a_1$, $a_2$, and $b$) reflects the time-varying contribution of the corresponding terms, thus enabling the full model to profile the transition of coded objects.

We compared the goodness-of-fit of the full model with that of the additive model, the multiplicative model, and a single cosine model (Eq. 1, 1st reach direction), by the standard of the population-averaged adjusted $R^2$ (adjusted $R^2$, a statistical method to compensate

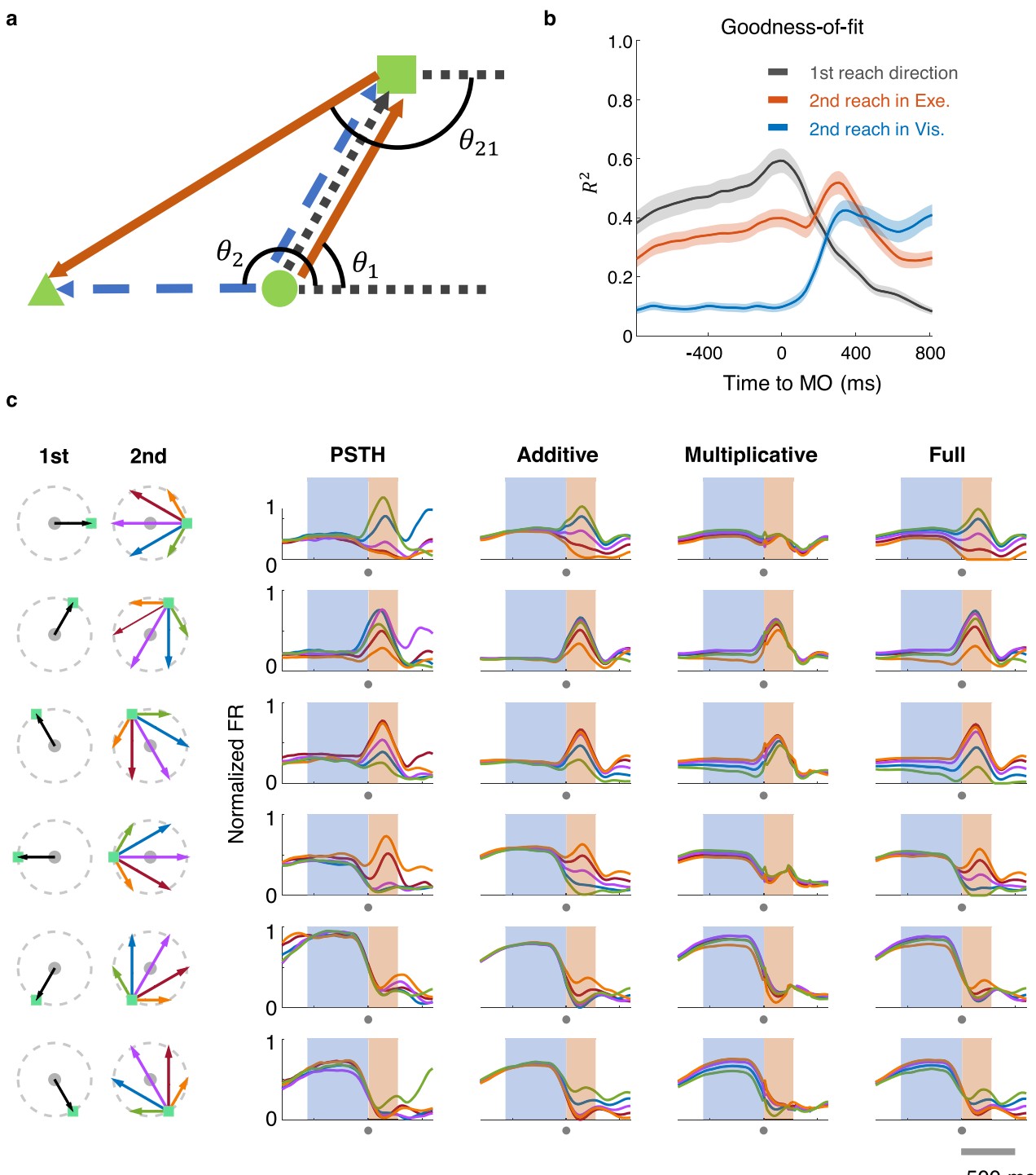

**Fig. 3 | Directional fitting of single neuron response. a** Visual vs. movement coordinates. Unlike the first reach, which was uniformly represented as a vector from the center to the square targets, the second reach might be encoded in either visual (from the center to the triangle, the blue vector, $\theta_2$) or motor (from the square to the triangle, the orange vector, $\theta_{21}$) coordinates. **b** The changing $R^2$ of cosine models in three coordinates was obtained with sliding windows (bin = 200 ms, step = 20 ms): first reach direction (gray) fits best before MO; second reach direction in execution coordinates (in Exe., red) fits well throughout the whole trial; second reach direction in visual coordinates (in Vis., blue) fits poorly

before MO. **c** The fitting result of an example neuron. Each row shows conditions with the same first reach (black arrow); the second reach is plotted in different colors (CW 60° in green, CW 120° in blue, 180° in purple, CCW 120° in red, CCW 60° in orange; here angle is according to the target locations in cue period). Four columns left to right are: Normalized data PSTHs; normalized firing rate reconstructed by the additive model, the multiplicative model, and the full model, respectively. All activity is aligned to MO (marked by the gray dots under the timeline, the blue shadow indicates 600 ms before MO, and the orange shadow indicates 250 ms after MO. Time window is −800–600 ms to MO).

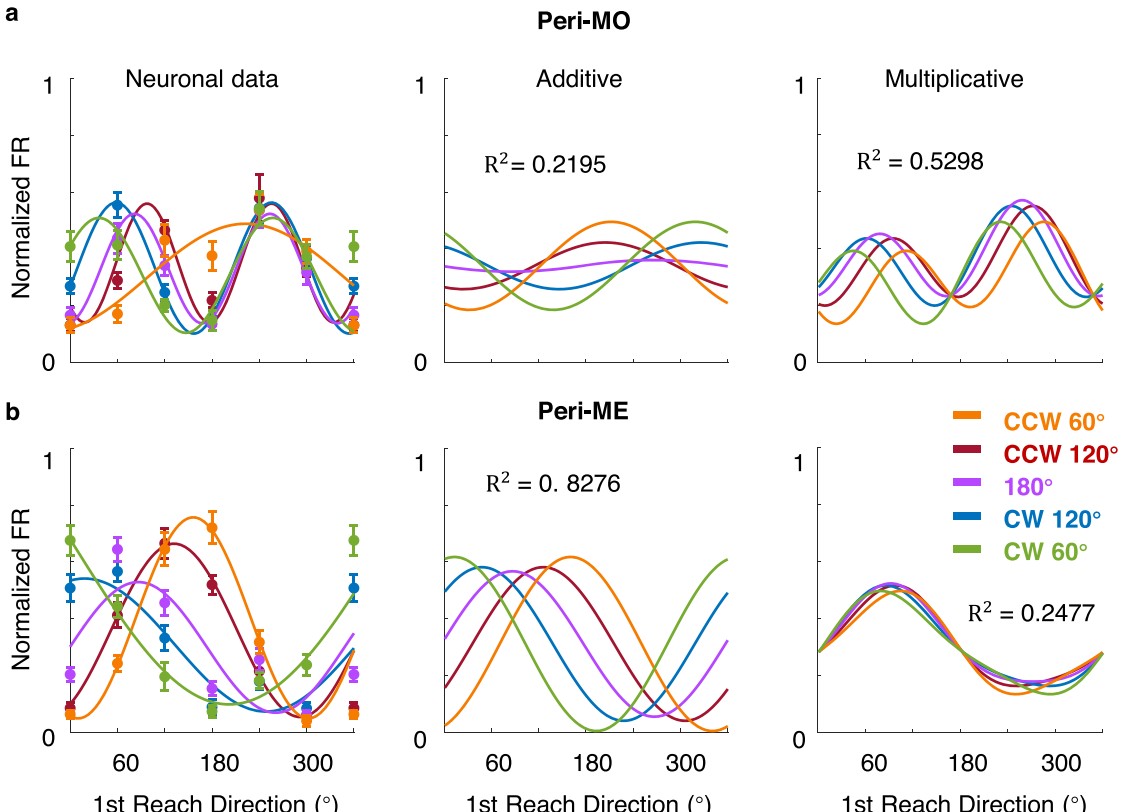

**Fig. 4 | Joint tunings of the example neuron around movement onset and end.**
**a** Directional tuning curves of the example cell in Fig. 3 were plotted around MO
(−100–100 ms to MO, peri-MO). Left: Normalized firing rates in DR were trial-
averaged and plotted in corresponding condition colors (DR trial number = 614, 20
trials per condition; mean ± 2 standard error). Tuning curves were fitted by Fourier
expansion separately. Middle: Tuning curves of firing rates reconstructed by the
addictive model. Right: Tuning curves of firing rates reconstructed by the multi-
plicative model. $R^2$ showed the goodness-of-fit of the model tuning curve. **b** Similar
to (**a**), directional tuning curves around ME (100–300 ms to MO, peri-ME).

for the difference in numbers of parameters between the full model
and other models, see Methods) for M1(Fig. 5a, array data from mon-
key C). The full model performed best; it was also able to describe
the tuning property of the example neuron throughout the whole
trial (Fig. 3c, Full). The goodness-of-fit for all models gradually
increased during preparation, and the multiplicative model was sig-
nificantly better than the additive model at MO (two-tailed Wilcoxon
signed-rank test, $p = 1.2e-05$). Nevertheless, the additive model per-
formed better after MO. Similar results were found in the standard
behavioral paradigm (CW or CCW 120° in Fig. 1) among all monkeys
(Fig. 5b, array data from monkey G; Supplementary Fig. 2, single-
electrode recording from monkeys B and C). The effect size $r$
(see Methods) also indicates there is a small to medium effect for the
multiplicative model during the preparatory period for each monkey
(Supplementary Fig. 3).

To scrutinize the changing encoding pattern, we plotted the
averaged absolute coefficients of the full model across time (Fig. 5,
right panel). The weights of the first reach and the multiplicative term
ramped up over the chance level (given by a permutation test, see
Methods) during preparation, whereas the additive weight remained at
the chance level in preparation and mainly increased after MO. This
contemporaneous activation of coefficients was similar to the situation
in the prefrontal cortex, where neurons were modulated by both
direction and sequence[31–33]. Similar dynamics were found in all mon-
keys (Supplementary Fig. 2), suggesting a common transition from a
gain-modulation interplay during motor preparation to a concurrent
coding during motor execution. This concurrence has been reported
by the previous study[27].

So far, we have analyzed the linear and nonlinear components
comprised in neural encoding for double-reach and their inter-
changeable predominance. The multiplicative joint coding, revealed
by the multiplicative model and validated by the multiplicative weight
in the full model, now becomes a key concern because it would be
apparently a unique signature of continuous motor sequences.

### Multiplicative coding embodied in initial states
According to our regression analyses, the multiplication of the tunings
corresponding to the first and second reaches could be intrinsic in
sequence-related preparatory activity. From the dynamical systems
perspective, preparatory activity would be set to a subspace optimal as
initial states to trigger motor generation[34]. We expected a spatially
inclusive distribution of initial states to accord with mathematical
multiplication.

To verify this hypothesis, we performed a supervised dimen-
sionality reduction procedure. Firstly, principal component analysis
(PCA) was applied to the preparatory neural activity during a period of
600 ms before GO. Next, Fisher's linear discriminant analysis (LDA)
was utilized to find the optimal discriminant projection in accordance
with tagged conditions[35]. In this PCA-LDA analysis, selected principal
components from PCA (the number was chosen by cross-validation)
were applied to LDA. Figure 6 shows the results from monkey C's array
data. We first analyzed neural activity in SR trials and built an SR sub-
space. Neural states clustered by conditions, as visualized in the 2D
projections found by LDA (Fig. 6a). Then, we projected both DR and SR
data onto the SR space and found that neural states of both DR and SR
trials clustered according to their first or only reach direction (Fig. 6b).

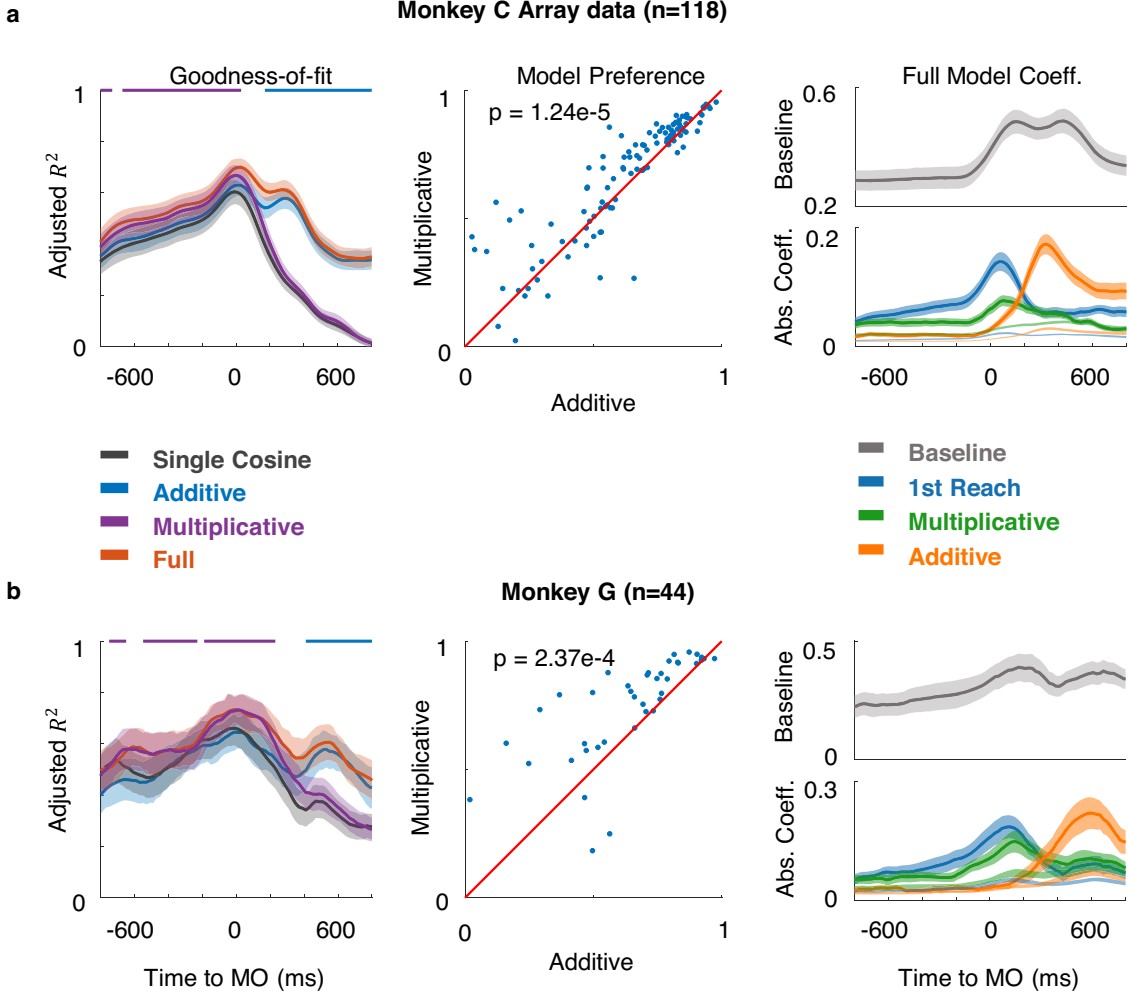

**Fig. 5 | Regression results of array datasets from two monkeys. a** Results of regression on M1 neurons in array dataset from monkey C. Left: Goodness-of-fit was evaluated with averaged adjusted $R^2$ for all fitting models in a 200-ms sliding window (± 2 standard error in shaded area). The upper line in purple shows the multiplicative model is significantly better ($p < 0.0005$, two-tailed Wilcoxon signed-rank test) than the additive model, while the blue line is vice versa. Middle: Scatters compared the goodness-of-fit at MO (−100–100 ms to MO) between the multiplicative and additive models, each dot represents the result of a neuron. Right: Mean of absolute coefficient values (± 2 standard error in error band). The coefficient weight of the permutation test was plotted in light shade as the chance level. **b** The results of array data from monkey G.

This suggests that despite the proposed sequence modulation in preparatory activity for single neurons, the neural population preserved a linear representation for the preceding movement. However, the explained variance was higher for SR than DR (For monkey C array, the explained variance of SR is 8.4%; that of DR is 6.9%. For monkey B, the explained variance of SR is 9.0%, and DR is 6.6%. For monkey C single electrode, the explained variance of SR is 6.2%, and DR is 5.6%. For monkey G, the explained variance of SR is 31.6%; that of DR is 27.5%.). To neutralize the tuning for the immediate movement, we used DR trials with the same 1st reach direction alone for the PCA-LDA analysis. Therefore, neural states could be projected onto dimensions maximizing the difference brought by the second reach directions (i.e., six clusters). We separately performed the LDA analysis for each of the six first movement directions, as shown in the subplots of Fig. 6c. The tenfold cross-validation accuracies are higher than the chance level in all directions (the tenfold cross-validation accuracies are 0.59, 0.64, 0.58, 0.60, 0.43, 0.43, from 0° to 300° subplots in Fig. 6c). There were great differences between SR (circles) and DR (other markers) clusters, indicating that the initial states for sequential movements were distinctive. Interestingly, in some conditions, DR trials obviously clustered in order from CW 60° to CCW 60°, and the CW and CCW states were located on both sides of the 180° states. This structural spatial distribution of LDA states is supported by Mahalanobis distances (Supplementary Fig. 5) between clusters in Fig. 6c, and may signify a condensation of subsequent movement information in the strong representation of occurrent movement. In addition, the results for other monkeys for the DR task showed a similar tendency (Supplementary Fig. 6).

We also examined the temporal dynamics of the information carried by the neural population by decoding both directions in DR trials. We trained LDA decoders in a sliding window (bin width = 300 ms, step = 20 ms) and plotted the tenfold cross-validation accuracy in Fig. 6d. Both movement directions can be decoded above the permutation level beginning with the preparatory period. The first reach can be perfectly decoded, while the second reach shows a lower accuracy, and is still ramping 400 ms after MO, which is close to the MO2. This result suggests that the planning of the first reach is earlier and more dominant than that of the second reach. The second reach information is implicitly embedded in the population response, beginning with the preparatory period and explicitly emerged during the execution period. This conclusion still holds true in a more rigorous decoding in the preparatory subspace (Supplementary Fig. 7). The temporal properties are displayed in the decoding results, aligned with the coefficients weights of the Full model in Fig. 5.

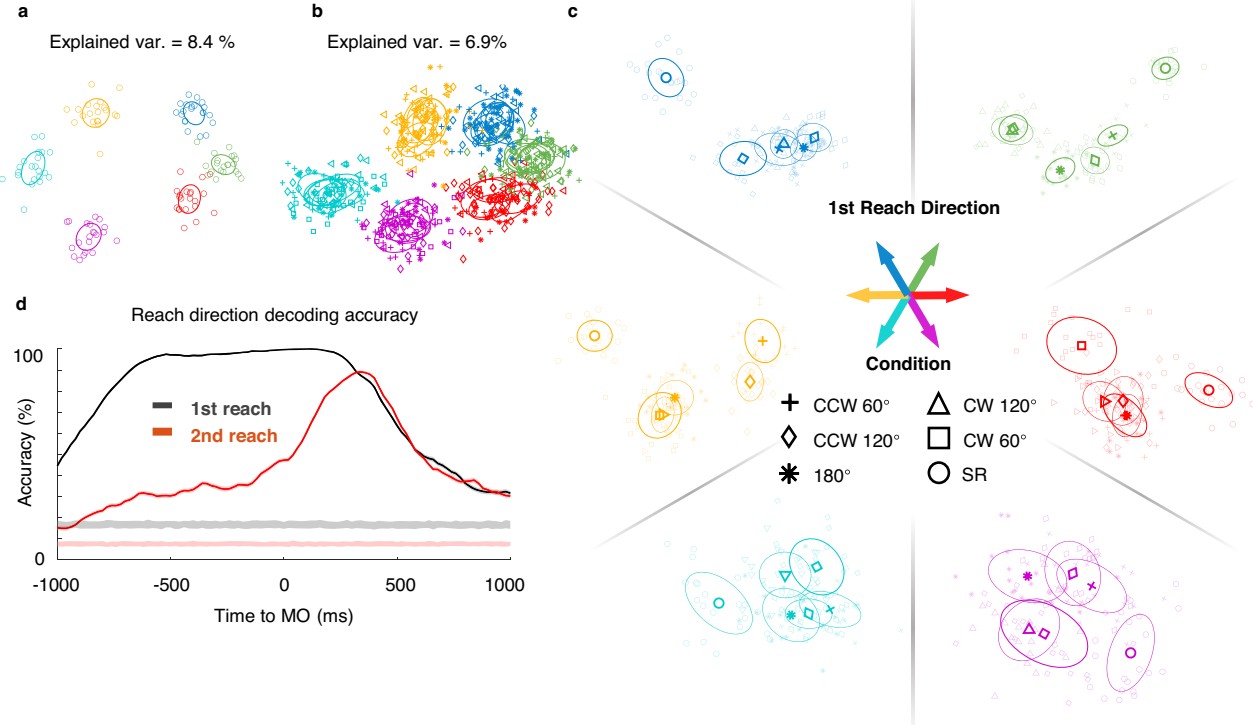

**Fig. 6 | Neural population embedded sequential modulation during the preparatory period. a** Projection on SR space. Neural states of SR trials ($n = 125$) were clearly clustered according to their reaching directions. **b** Neural states of DR trials ($n = 614$) also clustered into six groups according to their first reach direction when projected onto the SR space. The explained variances of the two dimensions were calculated. **c** LDA classified neural states of trials with the same first reach direction into clusters grouped by second reach directions, forming an initial state space for the subsequent movement. Colors indicate the first movement directions; DR trials are presented in the same color family of related SR trials. Markers indicate the second reaching direction. The ellipses show the covariance projection of related conditions. **d** Decoding accuracy for both reach directions in double-reach trials, using monkey C array DR trials. Trial conditions were shuffled 100 times to calculate permutation levels which are plotted in light shade curves.

## Multiplicative coding preserves linear readout of immediate reach

As several previous studies have pointed out[12,21,36,37], with a fixed linear readout like in the population vector (PV) or dimensional reduction method, the initial reach direction is captured during preparation, despite the sequence-related modulation at the single-neuron level (Supplementary Fig. 8). This is extremely interesting for the population vector, calculated from each neuron's PD, is expected to be sensitive to PD changes. But in sequential movement, this isn't observed. Some studies hypothesized this may be due to the neuronal connection coordination. We speculate that linear readout in sequential movements benefits from multiplicative joint coding, considering nonlinear mixed selectivity is believed to form high-dimensional neural representations that guarantee the linear readout of particular parameters[38].

To explore this, we simulated 200 neurons (see Methods[39]), defined their intrinsic PDs (the fixed parameter $\theta_{PD}$), and generated the single reach (SR) response with a single cosine model. Then generated the sequence modulated response for the initial reach under additive and multiplicative models. The generated responses are in an epoch of 600 ms from preparatory activity until the 1st reach end. Those additive and multiplicative neurons were regulated by a fixed second reach direction (CCW 120°). We present the responses of three sets of the response of one neuron with $\theta_{PD} = 336°$ in Fig. 7a. Obviously, the direction inducing the highest firing rate changed in additive and multiplicative modulated responses, compared to the "single cosine" response (Fig. 7b). We used the fixed $\theta_{PD}$ for the calculation of PV. Interestingly, PVs of the multiplicative responses correctly and stably pointed to the initial reach direction as in the SR responses, whereas PVs in the additive responses deviated from the desired direction

(Fig. 7c). These simulations show that multiplicative coding can preserve a robust fixed linear readout of immediate reach direction under sequence modulation.

## Multiplicative joint coding emerged in recurrent neural network (RNN) generating motor sequence

Due to their flexibility and time-varying characteristics, RNNs are increasingly welcomed as models matching a dynamical system[40–42]. To find out whether a dynamical system can also capture the subtle joint-coding rule found in the motor cortex, we trained an RNN model to perform the double-reach task.

The three-layer RNN received the Cartesian coordinates of two reaches and a Go signal as input (Fig. 8a). The target-relevant cues were presented simultaneously, though instructing sequential actions. In contrast to previous work in which RNNs were instructed to generate velocity[39] or EMG[43], our model was required to produce PV. This design was preferred for these reasons: first, the variables related to actual movement, like velocity and EMG, have to lag behind the neural activity due to transmission delay from cortex to muscle. In contrast, PV could be real-time, and thus reflect more temporal features; also, this design is consistent with our hypothesis that multiplicative joint coding benefits linear readout of movements (Fig. 7).

We trained 100 networks with different random seeds, these models performed well ($R^2 = 0.94 \pm 0.02$, mean squared error (MSE) = $0.05 \pm 0.02$, mean ± SD; see Methods). We selected one of them as an example ($R^2 = 0.97 \pm 0.01$, MSE = $0.03 \pm 0.02$). The model nodes exhibited comparable temporal dynamics with real neurons recorded in the present study. Here we show two example nodes under four specific conditions (Fig. 8b). For node 032, the two bumps of its response indicate that it is closely related to the ongoing movement, which is

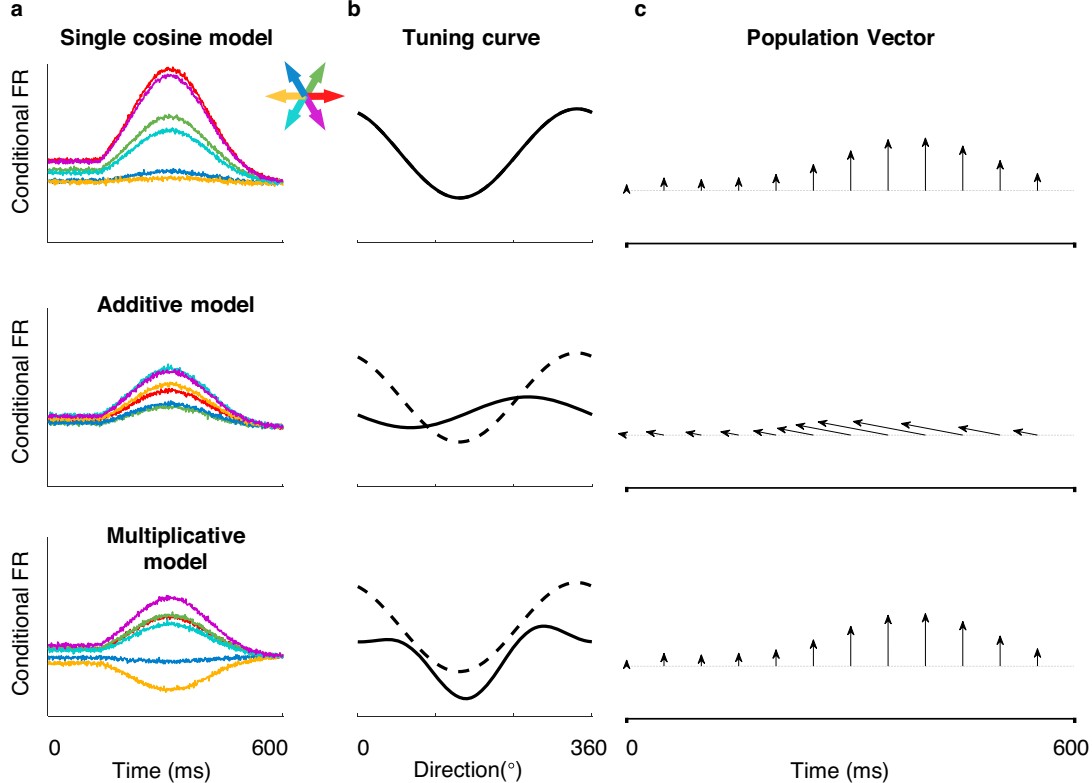

**Fig. 7 | Simulation of neural tunings on population vector during single and double reach. a** Example neurons of three simulated datasets. Averaged firing rates of different conditions (first reach directions) are shown in corresponding colors. These three example model neurons were simulated according to the single cosine, the additive, and the multiplicative models with the same preferred direction $\theta_{PD}$. **b** Directional tunning curves with (solid line) and without (dash line) modulation.

**c** Population vectors of three simulated datasets. Population vectors were calculated every 50 ms. The vector angle indicates the decoded reach direction while the correct reaching direction is upward. The population vector of the multiplicative dataset is pointed in the same direction as the PV of the single cosine dataset, while the PVs of the additive dataset shift away from the desired reaching direction.

typical for neurons in M1. It also seems to have 'direction selectivity', the only exceptive movement direction for the first reach (in cyan) induces obviously distinguished response. Node 012, however, only responds to the second reach, though maintaining input-driven dynamics during the preparatory period.

Observing richer preparatory dynamics than expected, we wondered whether the temporal dynamics of components corresponding to different movement courses were consistent between model and neural data. Therefore, we tested the 'full model' fitting on nodes of our models. The profile of regression coefficients of model nodes largely resembles that of real data (Supplementary Fig. 9a, Fréchet distance = 0.41 ± 0.04 compared with monkey C's array; In contrast, this for monkey C's single electrode recordings was 0.39, for monkey B was 0.56, for monkey G was 0.23. The average distance between monkey C's array and the permutation was 1.08; see Methods). As shown in Fig. 8c (Fréchet distance = 0.36), while the weight of the first reach peaks at MO and decays afterward, the weight of the additive term, which relates to the second reach, reaches its apex around MO2 with a smaller magnitude. During the preparation, the weight of the multiplicative term maintains a considerable influence. This suggests that the proposed multiplicative joint coding for sequential movement, here a double reach, also emerges in a dynamical system. The multiplicative coding even existed in a network with two triggers (Supplementary Fig. 10, the only difference in the training is that the Go signal pulses twice).

Furthermore, we found a similar initial state geometry in the neural latent space, by PCA. During preparation, the states with the same first reach direction were located nearby and then slightly

separated according to the second reach direction (Fig. 8d left). This structure was validated by the distance between states within the first-reach cluster, compared with that within DR conditions (Fig. 8e right and Supplementary Fig. 9b). During execution, the neural states cluster according to the ongoing reach direction (Supplementary Fig. 11). However, the causality is unclear given the perturbation on RNN via knocking down the nodes with different magnitude of coefficient weights failed to cause a significant difference in performance deficits among weights (Supplementary Fig. 9c).

## Discussion

In order to understand how the motor cortex generates motor programs for consecutive arm movement sequences, we recorded neuronal activity when monkeys performed double-reach directed at simultaneously cued memorized targets. We found that pre-movement activity carries sequence information in a heterogeneous manner. Regression analysis shows that neuronal tuning to first and second reaches can be well explained by multiplicative and additive models in the preparatory and execution periods, respectively. Dimensionality reduction analysis demonstrates that neural states during preparation sub-clustered according to the second reach within the optimal subspaces of the first reach. Simulation via model neurons points out the merit of multiplicative joint coding in maintaining robust linear readout for the ongoing movement direction. An RNN model trained for double-reach tasks can simulate the real encoding properties, which are marked by conspicuous nonlinearity. Taken together, these results suggest that the primate motor cortex is profoundly

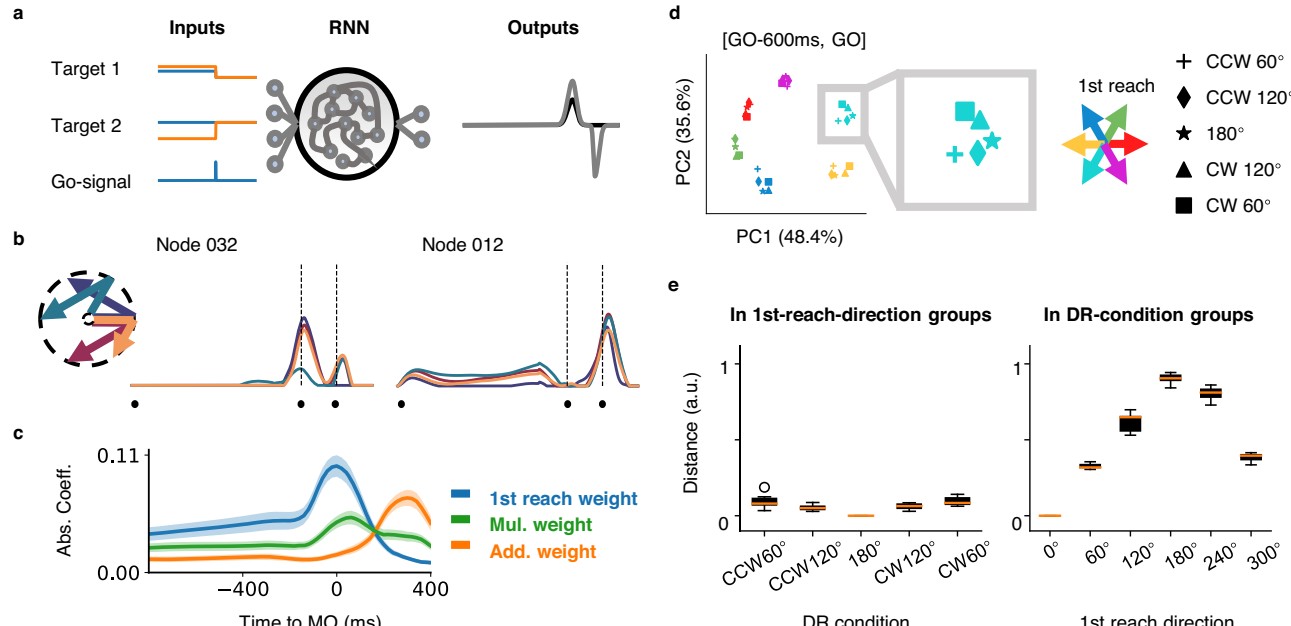

**Fig. 8 | Results of an example RNN model. a** Schematic of the RNN model. The RNN model consisted of an input layer, a hidden layer, and an output layer. The input layer received a signal for the position of two targets simultaneously, while the output layer produced a PV, whose magnitude reflects the degree of movement tendency for the neural population in the corresponding direction. **b** Response of two example nodes under four conditions. The selected conditions are represented in different colors, as shown on the left. The black dots denote the target on (TO), the first movement onset (MO), and the second movement onset (MO2), respectively. **c** Full model fitting result for RNN nodes. It turned out that the temporal patterns of the coefficients in this RNN model are comparable to those in

real neurons. The error band in this panel was plotted in mean ± 2 standard error. **d** The PCA neural states indicate initial states, during motor preparation. Colors indicate the first movement directions; DR trials are presented in the same color family as related SR trials. Markers indicate the second reaching direction. **e** Quantification of the clustering in (**d**). The normalized Euclidean distance of states under different DR conditions was calculated against the state under 180° condition, in the first reach-direction groups; the normalized Euclidean distance of states under different first reach conditions was calculated against the state under 0° condition, in the DR-condition groups.

involved in forming plans for multi-step movements. In addition, the transition between the newfound multiplicative joint coding and overlapped independent coding hints at a shifting neural encoding mechanism for motor sequences.

Previous studies have revealed that the motor cortex not only carries information regarding upcoming movements, but also reflects sensory and cognitive factors during both preparation and execution periods[6,9,12,16,17]. Nevertheless, how this "sequence selective" response reflects motor sequence has not yet been answered. A recent work following the dynamical systems perspective found that 'the preparatory subspace was occupied twice, once before each reach', thus suggested that each of the movement elements were encoded independently in the motor cortex rather than holistically. However, if individual movements were independently planned, the reaching error should accumulate, which has not yet been observed[44]. Furthermore, it was demonstrated that holistic planning might enhance motor learning, but such an effect would not occur when different follow-throughs were rehearsed individually[45]. This finding strongly suggests that sequential planning is associated with special neural states in preparation, in accordance with our findings. Although population response in the preparatory subspace showed little diversity before MO, the decoding accuracy of the second reach also ramped above the permutation accuracy before MO and rose to 50% after MO (Supplementary Fig. 7). Based on the results, we propose that elemental movements in sequential reach are modulated by the second movement in the sequence whose identity is fully defined and transferred into independent coding after MO. Also, unlike in the parietal cortex, neuronal activity in the motor cortex exhibits strong heterogeneity[46], which often comes from mixed selectivity of behavioral parameters and tuning dynamics[29,47–49]. Given these considerations and our results, we propose that elements in a consecutive movement sequence

should be interactively planned in a spatio-temporally coordinated manner beforehand.

As one of the cortical regions carrying much information regarding movement timing[10] and kinematics[11], the motor cortex presumably participates in encompassing and coordinating sequence components. In the present study, both reaching targets were turned off 400–800 ms before GO, encouraging the monkeys to plan the whole reaching sequence in the preparatory period. Our results revealed that neurons tended to jointly encode double reach in a multiplicative manner during preparation. The multiplicative model's performance degraded after MO, perhaps because joint coding mainly exists during preparation, as a reaching sequence is decomposed into motor elements, the lack of an additive term makes it incapable of capturing the parallel components after MO[27]. The concept of the multiplicative model originated from gain-modulation[50,51], and a work regarding the question of whether the neural response was constructed with nonlinear interactions between parameters, rather than their linear combination[29]. In the case of sequential movements, this issue becomes whether sequential elements are planned conjunctively or independently. As the primary nonlinear interaction, multiplication is a common form of gain-modulation that has been widely found in mixed selectivity[50,51]. This coding manner can provide new dimensions for motor preparation and learning[52], and according to our simulation, it can also consolidate the linear readout for impending movements. Because such mixed selectivity of parameters enlarges the neural space encoded by a certain number of neurons[35,38,52], the dimensionalities induced by multiplicative coding may perform as the null space of impending movement.

Although our analyses of joint coding are based on directional tuning, we did not mean to imply that the motor cortex exclusively encodes movement direction. Rather, we treated the directional

tuning as a marker of interaction, rooted in the heterogeneous neuronal response. Since the motor cortex is recognized to play a straightforward role in generating descending commands for muscle activity production[53,54], future studies should also take into account muscle activity to explain how joint coding benefits the generation of compound double reaches from a dynamical systems perspective[43]. However, it is a limitation of the present study that sEMG data were not sufficient to explore this issue.

Regarding joint coding embodied in the motor cortex as a key signature to encompass movement elements in the planning of consecutive sequences, we are not claiming that it seeds a neural dynamical system that can autonomously generate the entire motor sequence. Instead, sequential behavior emerges from a large brain network, including parietal-frontal circuits[9,55] and subcortical areas like the thalamus and basal ganglia[56]. Now that the dynamical evolution in the motor cortex necessarily relies on external inputs from other brain areas[57], an intriguing question is how intrinsic dynamics and external inputs interplay to generate a motor sequence, including the role of the proposed joint coding in the motor cortex. To go further, collective studies across multiple brain regions and experimental interventions are needed.

## Methods

### Experimental preparation

Three male rhesus macaques (monkeys B, C, and G, *Macaca mulatta*, 5–9 kg) were trained to perform a cohesive double-reach task (Fig. 1a). In each session, the monkey sat in a custom-designed primate chair. Stimuli were back projected onto a vertical touch screen (Elo Touchsystems, 19"; sampling at 100 Hz, spatial resolution <0.1 mm) -30 cm in front of the monkey. In the recording sessions using microelectrode arrays (Utah array, Blackrock), hand position was monitored optically via reflective markers attached to the wrist (Vicon Inc.), besides, acceleration and surface electromyography (sEMG) were recorded via a wireless sensor (Delsys Trigno Lab) attached to the targeted muscles. All procedures were in accordance with NIH guidelines and were approved by the Institutional Animal Care and Use Committee (IACUC) of the Institute of Neuroscience, CAS.

### Behavioral task

In addition to the standard version of the paradigm described in Results (Fig. 1a), to further examine the interaction between movement elements, we trained monkey C to perform an extended version of the task with multi-direction, in which the angle between the square and triangle could be 60° or 120° in both CW and CCW directions as well as 180°. This multi-direction task has 36 conditions in total (six SR and 30 DR).

### Data collection and analysis

For single-electrode recording, monkeys B and C were implanted with a standard recording cylinder (diameter = 19 mm) located over M1 and caudal PMd in the left hemisphere, guided by pre-scanned MRI and stereotactic coordinates. Recording sites are shown in Supplementary Fig. 1. Recordings were made using glass-coated tungsten electrodes (AlphaOmega, -1.5 MΩ impedance at 1 kHz). The activity was recorded online by an AlphaOmega Lab SNR system, and sampled at 44 kHz. After recordings, raw data were sorted offline according to an online template by Spike2 (Spike2 7.15, CED). For multi-electrode recording, monkeys G and C, respectively, were implanted with a 96-channel and two 128-channel Utah microelectrode arrays (Blackrock Microsystems, Salt Lake City, UT) in the motor cortex of the right hemisphere (Supplementary Fig. 1). Recording sites were located using MRI and cortex surface features. Array-recorded raw data were sorted offline by Wave_clus[58]. All monkeys were restricted to using the hand contralateral to the recorded hemisphere when performing the task. Data from monkey C were first obtained with a single microelectrode, and

subsequently from an array in the other hemisphere with a switch of hands.

In total, we collected 279 and 117 well-isolated units from monkeys B and C through single-electrode recording, respectively. Among these, 224 units from monkey B and 98 from monkey C with significant directional preference (One-way ANOVA, $p < 0.05$) in single reach were chosen for further analysis. For multi-electrode recording, we collected 169 and 63 well-isolated units of one session from monkeys C and G, respectively. Among these, 118 units from monkey C and 44 from monkey G with significant directional preference (one-way ANOVA, $p < 0.05$) were used. The selected neurons formed a three-dimensional NKT (N: neuron number, K: trial number, and T: spike time) dataset for regression and state-space analysis.

### Peri-stimulus time histograms (PSTHs)

For each unit, we calculated its PSTHs with time aligned to event markers such as the GO signal, the first/only movement onset (MO), the first/only movement end (ME), and the second movement onset (MO2). We defined MO as the moment when the monkey's hand left the touch screen and ME as the time when the monkey's hand touched the target on the screen. All firing rates were smoothed with a Gaussian kernel (SD = 20 ms). The mean standard error (mean SE) of the firing rate was estimated from ten bootstrap samples.

### Regression

We adopted the directional tuning model[1,29] to fit neural responses in the double-reach task. We fitted the normalized condition-averaged firing rates in a 200 ms sliding window with a 20 ms step (using Matlab function "fit' and 'fitnlm"). First, we fitted the double-reach data as follows:

$$FR = a\cos(\theta - \theta_{PD}) + c \qquad (6)$$

where $\theta$ is the movement direction, $\theta_{PD}$ is the PD, $a$ and $c$ denote regression coefficients. Both the first and the second reach direction were used for regression to see which direction is better represented at that time bin (Fig. 3b). Then we regressed double-reach data with the following models:

Additive model:

$$FR = a_1\cos(\theta_1 - \theta_{PD}) + a_2\cos(\theta_{21} - \theta_{PD}) + c \qquad (7)$$

Multiplicative model:

$$FR = a_1\cos(\theta_1 - \theta_{PD}) + b\cos(\theta_{21} - \theta_{PD})\cos(\theta_1 - \theta_{PD}) + c \qquad (8)$$

Full model:

$$FR = a_1\cos(\theta_1 - \theta_{PD}) + a_2\cos(\theta_{21} - \theta_{PD}) + b\cos(\theta_{21} - \theta_{PD}) \\ \cos(\theta_1 - \theta_{PD}) + c \qquad (9)$$

where $a_1, a_2, b, c$ are regression coefficients, $\theta_1$ is the first movement direction, $\theta_{21}$ is the second movement direction from the first reach endpoint, $\theta_{PD}$ is the preferred direction.

Note that both the additive and multiplicative models have four coefficients, while the full model has five. To compensate for this difference, we use the adjusted $R^2$ rather than actual $R^2$,

$$R_{adj}^2 = 1 - \left(\frac{n-1}{n-p}\right)\frac{SSE}{SST} \qquad (10)$$

where SSE is the sum of squared error, SST is the sum of squared total, $n$ is the number of observations, and $p$ is the number of regression coefficients. Because actual $R^2$ likely increases with added predictor

variables in the regression model, the adjusted $R^2$ adjusts for the number of predictor variables in the model. This makes it more useful for comparing models with a different number of predictors.

We also compared the goodness-of-fit between the multiplicative and additive models using the Wilcoxon signed-rank test. We plot a line (purple for the multiplicative model, blue for the additive model) when one is significantly ($p < 0.0005$) better than the other.

In addition, we calculated the effect size $r = Z/\sqrt{n}$ using function "wilcoxonPairedR" in package "rcompanion" of R (Mangiafico, S.S. 2016. Summary and Analysis of Extension Program Evaluation in R, version 1.19.10. rcompanion.org/handbook/[59]). The $r$ value could be interpreted as a small effect in 0.1–0.4, a medium effect in 0.4–0.6, and a large effect $\geq 0.6$.

To get the chance levels of each coefficient and to reflect the effect of modulation, we performed a permutation test with 1000 repetitions separately for the coefficient of the first reach, multiplicative term, and additive term in reference of Sober and Sabes[60].

## PCA-LDA analysis for neural states

NKT datasets were used in this analysis. Neuronal firing rates were calculated with a 300 ms bin width ($T = 2$) and normalized by Z-score (MATLAB function "*zscore*") to avoid bias from high firing rate neurons. NKT data were reshaped into K × NT, where K is trial number, N is neuron number, and T is bin number. For building the SR space (in Fig. 6a, b), we applied PCA only on the SR trials to get the principal component coefficients (and for DR space in Fig. 6c, we applied PCA only on the relevant trials). The data was reduced to $K \times P$. The number of PCs, $P$, was chosen by tenfold cross-validation to avoid overfitting. This step also helped avoid singular matrices for LDA and reduced data noise[35]. Then we ran LDA to project the $P$-dimensional matrix onto a $C$-dimensional space, where $C$ is the number of trial conditions. LDA can find axes that best separate the categories. After this, we applied QR decomposition to get the orthonormal basis for the neural state space[61]. Each trial was finally described by $C - 1$ components derived from selected neural activity. We chose the first two components covering the largest variance to plot the 2D projection of trial data and the ellipse of covariance; each data point represented the neural state in a trial.

## Simulation of population vector in sequential reach

We adopted the simulation method of ref. 39 to generate surrogate data based on single cosine, additive, and multiplicative models. The preparatory and peri-movement activity were simulated with 200 neurons in six directions. The averaged neuronal firing rate $f_{n,c}$ for neuron $n$, in condition $c$, at time $t$ is given by

$$f_{n,c}(t,\tau_n,\sigma) = \begin{cases} b_{n,c} e^{-\frac{(t-\tau_n-\mu_0)^2}{2\sigma^2}} + \varepsilon, t \geq \tau_n \\ \varphi b_{n,c} + \varepsilon, t < \tau_n \end{cases} \quad (11)$$

where $\sigma$ is the duration parameter, $\tau_n$ is the response latency of each neuron (normally distributed), $\varphi$ is the preparatory activity amplitude constant fixed at 0.2, $\mu_0$ is constant given by $\mu_0 = \sigma\sqrt{-2\ln\varphi}$, and $\varepsilon$ is random noise (SD = 0.01). $b_{n,c}$ is the gain for neuronal condition preference. For data of the cosine model, which is expected to mimic neuronal activity in SR trials, $b_{n,c}$ is simply tuned to reach directions as

$$b_{n,c} = \frac{1 + \cos(\theta_1 - \theta_{PD})}{2} \quad (12)$$

The additive surrogate data were based on the parallel coding hypothesis that sequential movements are planned independently

with the overlap in the peri-movement period; $b_{n,c}$ is given by

$$b_{n,c} = \frac{1 + \cos(\theta_1 - \theta_{PD}) + \cos(\theta_{21} - \theta_{PD})}{3} \quad (13)$$

The multiplicative surrogate data were based on the gain-modulation hypothesis, the interaction of both movement directions in sequential reach contributed to the neuronal response,

$$b_{n,c} = \frac{1 + \cos(\theta_1 - \theta_{PD}) + \cos(\theta_1 - \theta_{PD})\cos(\theta_{21} - \theta_{PD})}{3} \quad (14)$$

For the above definitions, $\theta_1$ is the first movement direction, $\theta_{21}$ is the second movement direction relative to the 1st movement end-point, and $\theta_{PD}$ is the preferred direction.

## Model training

Our RNN model was designed to simulate the situation where double reach was accomplished by a pure dynamical system. The input was movement direction for two sequential reaches, in the form of 2D Cartesian coordinates $[\cos(\theta_1), \sin(\theta_1); \cos(\theta_2), \sin(\theta_2)]$, where $\theta_1$ and $\theta_2$ represent the first and relative second movement directions, respectively. Because the model was built to generate population vectors (PVs), we constructed "desired PVs" instead of using real data for generality. The output was read out as $[r\cos(\theta), r\sin(\theta)]$, where $\theta$ is the present movement direction, and $r$ reflects the intensity of integrated response for the population. We used Gaussian functions to emulate the time-varying magnitude. To ensure the trend at critical time markers was similar to the actual situation, we separated the two-peak PV profile into four sections: from GO to MO, from MO to the first touch, from MO2-50 ms to MO2, and from MO2 to the second touch, and spliced them together after respective optimization and normalization. We used 36 standard conditions in training and validation, as mentioned in the multi-direction task.

The nodes in the RNN model were evolved according to a standard continuous dynamical equation[39]:

$$\tau \dot{x}_i(t) = -x_i + \sum_{k=1}^{N} J_{ik} r_k(t) + \sum_{k=1}^{I} B_{ik} u_k(t) \quad (15)$$

where $\tau$ is a time constant, $N$ is the number of network nodes, and $I$ is the number of the inputs. The activity of nodes is represented by $x$, whose firing rates are determined by

$$r = \begin{cases} 0, x < 0 \\ \tanh(x), x \geq 0 \end{cases} \quad (16)$$

The output was read out linearly as:

$$z_i = \sum_{k=1}^{N} W_{ik} r_k(t) \quad (17)$$

where $z$ represents the two PV readouts ($i = 1,2$). In this model, the connection weight among nodes is denoted by matrix $J$, the connectivity between hidden nodes and input $u(t)$ is defined by matrix $B$, and the weight matrix between hidden nodes and output is $W$.

The size of our RNN was fixed at 200. We initialized the internal connection matrix $J$ to be normally randomized(mean = 0, SD = $g/\sqrt{N}$; $g = 1.5$), the input connection matrix $B$ and output connection matrix $W$ to be both uniformly randomized (between $-10^{-3}g/\sqrt{N}$ and $10^{-3}g/\sqrt{N}$), and chose a time constant $\tau = 50$ ms in the light of previous work[39,62].

All three weights were adjustable and optimized during training. We used the summation of the error function added with a regularity term as a cost function[43]. The error function was the mean squared error between the model output and the desired PV. The regularity

term penalized the magnitude of the squared firing rate averaged by neuron size, time bins, and condition numbers. The training was finished with PyTorch, and the weights were optimized by Adam (adaptive moment estimation).

To compare the pattern of coefficients, we visualized the three time-varying coefficients as a normalized 3D trajectory, and calculated the Fréchet distance between trajectories of different sessions or monkeys.

For the perturbation, the high group included those nodes with the highest 10% weights, while the low group included those nodes with the lowest 10% weights, and the random group randomly picked 10% of all activated nodes. Each time, given a trained network, ten nodes from a certain group were randomly selected and then knocked down (all relevant connections set to zero). Then we tested the modified network with the same validation set to see the perturbed performance. We bootstrapped this 100 times and got the statistics.

### Reporting summary

Further information on research design is available in the Nature Portfolio Reporting Summary linked to this article.

## Data availability

The data used in this study are available at https://doi.org/10.5281/zenodo.10637304 Source data are provided with this paper.

## Code availability

The custom-written codes used to analyze data from this study are available at https://github.com/Twwang13/Double_Reach.

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

## Acknowledgements

We thank Y. Guo, J. Malpeli, Q. Wang, C. Zheng, and R. Zheng for helpful comments and discussions; C. Guan and L. Wang for veterinary assistance; J Li for creating key monkey cartoons; P. Ding for administrative support. This work was supported by the National Key R&D Program (2017YFA0701102 and 2020YFB1313400), the National Science Foundation of China (31871047 and 31671075), Shanghai Municipal Science and Technology (18JC1415100 and 2021SHZDZX), and Strategic Priority Research Program of Chinese Academy of Science (Grant No. XDB32040100).

## Author contributions

T. Wang, Y. Zhang, and H. Cui designed the experiment, T. Wang and Y. Zhang collected the data, T. Wang analyzed the data, Y. Chen built a computational model, and T. Wang, Y. Chen, and H. Cui prepared the manuscript.

## Competing interests

The authors declare no competing interests.
