## [Peer Review File · Nature Communications]

Multiplicative Joint Coding in Preparatory Activity for Reaching Sequence in Macaque Motor CortexEditorial Note: This manuscript has been previously reviewed at another journal that is not operating a transparent peer review scheme. This document only contains reviewer comments and rebuttal letters for versions considered at *Nature Communications*.

REVIEWERS' COMMENTS

Reviewer #1 (Remarks to the Author):

Thank you for the thorough revisions to the paper, which I feel have improved the manuscript. I have some additional comments, both related to my previous remarks and some other aspects of the manuscript.

Comments regarding the previous round of revisions (I appear listed as Reviewer #1):

- Comment 1: Figure R1 is nice; I'd encourage the authors to include it as a main figure in the paper.

Figure R2 is also interesting, but the authors need to do one additional check. As it stands, this analysis assumes that the preparatory dimensions during the single reach are preserved during double reaches (including during reach 2); this should be verified, e.g., using the same method from Elsayed, Lara et al (Nature Comms 2016) that they used in Figure R2b, or principal angles (e.g., Gallego et al Nature Comms 2018) to ask whether the preparatory subspace is the same across single and double reaches (and across reaches in the double reach condition). I'd also strongly advise the authors to include this figure as a Supplementary figure in the paper.

- Figure R3 is very interesting and I'm glad to see it in the paper. But it also raises an important question that needs to be investigated. The authors argue for a multiplicative coding of both reaches during preparation, yet their ability to decode the second reach prior movement onset (MO) plateaus at around 40%. How does this fit with their proposal of joint multiplicative coding of the reaching sequence? Because to me this indicates that there's some shadow of the second movement in the sequence but it's identity isn't fully defined —although perhaps can be inferred when using more complex decoder models.

(Also, the authors should indicate the chance level)

- Figure R4: This is a nice analysis but you should make a more rigorous quantification (e.g., distances across representations of the second reaches for each first reach) and run it across multiple networks seeds (parameter combinations), and report the outcome with appropriate statistics

General comments on manuscript:

1. All figures in the manuscript should indicate not only the onset of the first movement (denoted by the authors as MO), but also when it ends (ME) and when movement two begins (MO2); otherwise one cannot understand their relationship with the sequential movements. Related, the memory period was variable, so I was expecting to see the usual blank space separating epochs when trial-averaging (e.g., in Figure 3 the time window starts 800 ms before MO but if I understood correctly the shortest memory period is 400 ms). What am I missing?

2. Figure 5 middle panel and the associated supplementary figure are very important for the authors' argument since they directly compare the additive and multiplicative models. My understanding is that they have only done it for a particular epoch (peri-movement activity around movement 1, MO), but the authors should also look at data for different epochs (e.g., preparation, around the onset of the second movement) and at least include it in the Supplement.

3. Apologies if I missed this in the method but in Figure 6c, did the authors use an LDA separately for each second reach, or is it the same across all six initial reaches?

4. I think the paper has improved since the last revision, but I have trouble understanding the section "Multiplicative coding preserves linear readout of immediate reach" and Figure 7. Personally, I don't think it adds much to the paper, so my suggestion would be to either remove it from the main text, or if not, explain it better (e.g., what does the angle of the population vector in Fig 7c represent?).

5. The RNN part of the manuscript has improved but the authors should: a) perform additional checks as to whether their key result generalise across more than one combination of parameters; b) take better advantage of the models.

Regarding (a), it is well known the way inputs are given to RNNs shape their internal dynamics. Thus, I was wondering if having two trigger signals rather than a single go cue would still reproduce their experimental results (note that one could assume, like Zimnik & Churchland, that there is a trigger signal coming from somewhere else within the brain that triggers each movement, hence the suggestion of two trigger signals instead of one). Doing this comparison would be very interesting since the authors can compare both sets of models to the actual data and gain more insight into the data.

Regarding (b), the authors mention in their reply that they could perturb the model in future studies, which I definitely think they should do it in this manuscript to provide causal evidence that manipulating the pre-movement (pre-MO) information about the second target impairs the execution of the second reach. The authors should again check whether this is true for a broad combination of model choices (e.g., having one or two trigger signals).

Finally, the authors should include the LDA figure that they made for the response letter (R4), and run all the RNN results across many seeds and report statistics.

Specific remarks:

- The authors explain in the response to comment 2 that they assume that “the neuron has the same θ_{pd} in a certain time bin across condition as a latent property (...)”. This is a nice explanation of their single neuron model that they should consider adding to the paper.

- Figure 5 should include results from multiple monkeys (currently only shown in the supplement). Also the right hand side traces should have error bars, and the caption is lacking in details (as are a few others).

- In the abstract and the discussion, the authors argue that their data supports a multiplicative model during planning, in which both the first and second reaches are encoded by the neural population. However, their ability to record the second reach during the planning epoch and early during movement is not that far from chance and far away from their ability to decode the first reach. This is a crucial point that I think should be addressed and the manuscript edited accordingly.

Reviewer #2 (Remarks to the Author):

In their second review the authors made a strong effort to respond to the large set of detailed comments made by reviewer 1. In general, they structured a comprehensive and compelling set of answers regarding the validity of the multiplicative representation of two reaching direction in the primate motor cortex. After the reviews, the paper is making a large contribution to the knowledge of sequence representation in the voluntary motor system of the monkey.

REVIEWERS' COMMENTS

Reviewer #1 (Remarks to the Author):

Thank you for the thorough revisions to the paper, which I feel have improved the manuscript. I have some additional comments, both related to my previous remarks and some other aspects of the manuscript.

Comments regarding the previous round of revisions (I appear listed as Reviewer #1):

- Comment 1: Figure R1 is nice; I'd encourage the authors to include it as a main figure in the paper.

Reply:

As suggested, we have included Fig.R1 in new Fig.3.

Figure R2 is also interesting, but the authors need to do one additional check. As it stands, this analysis assumes that the preparatory dimensions during the single reach are preserved during double reaches (including during reach 2); this should be verified, e.g., using the same method from Elsayed, Lara et al (Nature Comms 2016) that they used in Figure R2b, or principal angles (e.g., Gallego et al Nature Comms 2018) to ask whether the preparatory subspace is the same across single and double reaches (and across reaches in the double reach condition). I'd also strongly advise the authors to include this figure as a Supplementary figure in the paper.

Reply:

Thanks for the advice. We agree that the preparatory dimensions during the single reach are preserved during double reaches, which is also supported by the results in Fig. 6a&b. In addition, we have put this figure in the supplementary Fig. 7 as following:

Line 381-386

“Although population response in the preparatory subspace showed little diversity before MO, the decoding accuracy of 2nd reach also ramped above the permutation accuracy before MO and rose to 50% after MO (Supplementary Fig. 7). Based on the results, we propose that elemental movements in sequential reach are modulated by the second movement in the sequence whose identity is fully defined and transferred into independent coding after MO.”

- Figure R3 is very interesting and I'm glad to see it in the paper. But it also raises an important question that needs to be investigated. The authors argue for a multiplicative coding of both reaches during preparation, yet their ability to decode the second reach prior movement onset (MO) plateaus at around 40 %. How does this fit with their proposal of joint multiplicative coding of the reaching sequence? Because to me this indicates that there's some shadow of the second movement in the sequence but its identity isn't fully defined—although perhaps can be inferred when using more complex decoder models.

(Also, the authors should indicate the chance level)

Reply:

We totally agree with this comment and the description has been refined as Line 274-284:

“We also examined the temporal dynamics of the information carried by the neural population by decoding both directions in DR trials. We trained LDA decoders in a sliding window (width = 300 ms, step = 20 ms) and plotted the ten-fold cross-validation accuracy in Fig. 6d. Both movement directions can be decoded above the permutation level beginning with preparatory period. The 1st reach can be perfectly decoded while the 2nd reach shows a lower accuracy, and is still ramping 400ms after MO, which is close to the MO2. This result suggests that the planning of the 1st reach is earlier and more dominant than that of the 2nd reach. The 2nd reach information is implicitly embedded in the population response beginning with preparatory period and explicitly emerged during execution period. This conclusion still holds true in a more rigorous decoding in the preparatory subspace (Supplementary Fig. 7). The temporal properties are displayed in the decoding results, aligned with the coefficients weights of the Full model in Fig.5.”

- Figure R4: This is a nice analysis but you should make a more rigorous quantification (e.g., distances across representations of the second reaches for each first reach) and run it across multiple networks seeds (parameter combinations), and report the outcome with appropriate statistics

Reply:

Thank you for the advice. We’ve calculated the normalized Euclidean distance between representations and ran it across 100 network seeds to get a statistical report. In the revised manuscript, we’ve included the result of an example network (Fig. 8) and the statistical result (Supplementary Fig. 9 and Supplementary Fig. 11).

General comments on manuscript:

1. All figures in the manuscript should indicate not only the onset of the first movement (denoted by the authors as MO), but also when it ends (ME) and when movement two begins (MO2); otherwise one cannot understand their relationship with the sequential movements. Related, the memory period was variable, so I was expecting to see the usual blank space separating epochs when trial-averaging (e.g., in Figure 3 the time window starts 800 ms before MO but if I understood correctly the shortest memory period is 400 ms). What am I missing?

Reply:

Sorry for not making it clearer. Indeed, the delay from Cue on to MO was at least 800 ms (400ms of cue period + 400-800ms of memory period = 800-1200 ms). In addition, since our analyses focused on peri-MO periods, separated epoch representation wouldn’t provide additional insight. We have clarified this in the behavioral section (Line 80-84):

“... another green dot was presented as a reaching goal for 400 ms (cue period) at one of the six corners of a regular hexagon (i.e., at directions of 0°, 60°, 120°, 180°, 240°, or 300°). After the

peripheral cue was extinguished, there was a memory period of 400-800 ms. Thus, the total delay from Cue on to GO was 800-1200 ms.”.

2. Figure 5 middle panel and the associated supplementary figure are very important for the authors’ argument since they directly compare the additive and multiplicative models. My understanding is that they have only done it for a particular epoch (peri-movement activity around movement 1, MO), but the authors should also look at data for different epochs (e.g., preparation, around the onset of the second movement) and at least include it in the Supplement.

Reply:

As suggested, we have done the statistical analysis for all epochs. To show the comparison result, we’ve plotted upper lines in the subplots in left pane to indicate the significance. The detailed description reads as (Line 799-801):

“The upper line in purple showed the multiplicative model is significantly better ($p < 0.0005$, two-tailed Wilcoxon signed rank test) than the additive model, while blue line vice versa.”

As the reviewer suggested, we also added the scatters (like the Fig.5 middle panel) of other time of all monkeys in the supplementary Supplementary Fig. 4.

3. Apologies if I missed this in the method but in Figure 6c, did the authors use an LDA separately for each second reach, or is it the same across all six initial reaches?

Reply:

We use the LDA separately according to the 1st reach directions of double reach trials, therefore it’s not same across all six initial reaches. Now this part reads (Line 262-263):

“We separately performed the LDA analysis for each of the six 1st movement directions, as shown in the subplots of Fig. 6c.”

4. I think the paper has improved since the last revision, but I have trouble understanding the section “Multiplicative coding preserves linear readout of immediate reach” and Figure 7. Personally, I don’t think it adds much to the paper, so my suggestion would be to either remove it from the main text, or if not, explain it better (e.g., what does the angle of the population vector in Fig 7c represent?).

Reply:

Thanks for the advice. The linear readout robustness is a central advantage of mixed selectivity like multiplicative coding. We highly understand that it is not a novel phenomenon reported in motor cortex, but it explained the reason why we use PV as the output of RNN and may have provided support for the function of mixed selectivity. As the reviewer mentioned, we find the question discussed here is not clarified and some part of the method maybe confusing. We refined the section and now it reads (Line 288-309):

“As several previous studies have pointed out^{12, 21, 37, 38}, with a fixed linear readout like in the

PV or dimensional reduction method, the initial reach direction is captured during preparation, despite the sequence-related modulation at the single-neuron level (Supplementary Fig. 8). This is extremely interesting for the population vector, calculated from each neuron's preferred direction (PD), is expected to be sensitive to PD changes. But in sequential movement, this isn't observed. Some studies hypothesized this may be due to the neuronal connection coordination. We speculate that linear readout in sequential movements benefits from multiplicative joint coding, considering nonlinear mixed selectivity is believed to form high-dimensional neural representations that guarantee the linear readout of particular parameters³⁹.

To explore this, we simulated 200 neurons (see Methods 40), defined their intrinsic PDs (the fixed parameter θ_{PD}), and generated the single reach (SR) response with single cosine model. Then generated the sequence modulated response for the initial reach under additive and multiplicative models. The generated responses are in an epoch of 600 ms from preparatory activity until the 1st reach end. Those additive and multiplicative neurons were regulated by a fixed 2nd reach direction (CCW 120°). We present the responses of three sets of response of one neuron with $\theta_{PD} = 336$ in Fig. 7a. Obviously, the direction inducing the highest firing rate changed in additive and multiplicative modulated responses, compared to the 'single cosine' response (Fig.7b). We used the fixed θ_{PD} for the calculation of PV. Interestingly, PVs of the multiplicative responses correctly and stably pointed to the initial reach direction as in the SR responses, whereas PVs in the additive responses deviated from the desired direction (Fig.7c). These simulations show that multiplicative coding can preserve a robust fixed linear readout of immediate reach direction under sequence modulation.”

5. The RNN part of the manuscript has improved but the authors should: a) perform additional checks as to whether their key result generalise across more than one combination of parameters; b) take better advantage of the models.

Regarding (a), it is well known the way inputs are given to RNNs shape their internal dynamics. Thus, I was wondering if having two trigger signals rather than a single go cue would still reproduce their experimental results (note that one could assume, like Zinnik & Churchland, that there is a trigger signal coming from somewhere else within the brain that triggers each movement, hence the suggestion of two trigger signals instead of one). Doing this comparison would be very interesting since the authors can compare both sets of models to the actual data and gain more insight into the data.

Regarding (b), the authors mention in their reply that they could perturb the model in future studies, which I definitely think they should do it in this manuscript to provide causal evidence that manipulating the pre-movement (pre-MO) information about the second target impairs the execution of the second reach. The authors should again check whether this is true for a broad combination of model choices (e.g., having one or two trigger signals).

Finally, the authors should include the LDA figure that they made for the response letter (R4), and run all the RNN results across many seeds and report statistics.

Reply:

We appreciated these suggestions. Regarding (a), the proposed comparison is really interesting but we are afraid that it may diverge from the content of this article. Therefore, we tried a network with two trigger signals as an extension (Supplementary Fig. 10), in moderated discussion. Regarding (b), we tried the perturbation. Specifically, we chose to knock down those nodes with certain magnitude of coefficient weights rather than manipulate the pre-movement information. The perturbation was implemented across 100 network seeds as mentioned above.

Fig. R1 The perturbation results across 100 models.

The high group included those nodes with the highest 10% weights while the low group included those nodes with the lowest 10% nonzero weights. The random group included all nodes with non-zero activity. Each time, given a trained network, ten nodes from a certain group were randomly knocked down (all relevant connection set to zero). Then, the modified network was tested with the same validation set. We bootstrapped this for 100 times and obtained the statistics. The perturbed performances during whole movement, 1st reach, and 2nd reach were shown separately.

Even though we found those nodes with high multiplicative weights significantly prior than those nodes with low weights ($p < 0.001$) and random multiplicative weights ($p < 0.001$), the results were elusive and scarcely yielded concise insights. Therefore, we decided put this in Supplementary Fig. 9c.

Specific remarks:

- The authors explain in the response to comment 2 that they assume that “the neuron has the same θ_{pd} in a certain time bin across condition as a latent property (...)”. This is a nice explanation of their single neuron model that they should consider adding to the paper.

Reply:

Thanks for the advice. We added this explanation in Line 194.

“Here, we assume the neuron has the same θ_{pd} in a certain time bin across conditions in each model.”

- Figure 5 should include results from multiple monkeys (currently only shown in the supplement). Also the right hand side traces should have error bars, and the caption is lacking in details (as are a few others).

Reply:

As suggested, we've included all the array dataset of monkey G in the Fig 5. For the sake of simplicity and convenience of layout, we choose to put the single electrode result in the supplementary Fig. 2.

The shaded areas of the right-hand side traces are the error bar indicating the 95% intervals of the average coefficients.

We have refined the captions as follow Line 797-804:

“Figure 5 Regression results of array datasets from two monkeys

a. Results of regression on M1 neurons in array dataset from monkey C. Left: Goodness-of-fit was evaluated with averaged adjusted R2 for all fitting models in a 200-ms sliding window (± 2 standard error in shaded area). The upper line in purple showed the multiplicative model is significantly better ($p < 0.0005$, two-tailed Wilcoxon signed rank test) than the additive model, while blue line vice versa. Middle: Scatters compared the goodness-of-fit at MO (-100~100 ms to MO) between the multiplicative and additive models, each dot represents the result of a neuron. Right: Mean of absolute coefficient values (± 2 standard error in shaded area). The coefficient weight of permutation test was plotted in light shade as the chance level. b. The results of array data from monkey G.”

- In the abstract and the discussion, the authors argue that their data supports a multiplicative model during planning, in which both the first and second reaches are encoded by the neural population. However, their ability to record the second reach during the planning epoch and early during movement is not that far from chance and far away from their ability to decode the first reach. This is a crucial point that I think should be addressed and the manuscript edited accordingly.

Reply:

Thank you for the suggestions! In the revised manuscript, we've emphasized the difference in decoding accuracy and refined the section about the decoding result in Line 274-284:

“We also examined the temporal dynamics of the information carried by the neural population by decoding both directions in DR trials. We trained LDA decoders in a sliding window (bin width = 300 ms, step = 20 ms) and plotted the ten-fold cross-validation accuracy in Fig. 6d. Both movement directions can be decoded above the permutation level beginning with preparatory period. The 1st reach can be perfectly decoded while the 2nd reach shows a lower accuracy, and is still ramping 400ms after MO, which is close to the MO2. This result suggests that the planning of the 1st reach is earlier and more dominant than that of the 2nd reach. The 2nd reach information is implicitly embedded in the population response beginning with preparatory period and explicitly emerged during execution period. This conclusion still holds true in a more rigorous decoding in the preparatory subspace (Supplementary Fig. 7). The temporal properties are displayed in the decoding results,

aligned with the coefficients weights of the Full model in Fig.5.”

Reviewer #2 (Remarks to the Author):

In their second review the authors made a strong effort to respond to the large set of detailed comments made by reviewer 1. In general, they structured a comprehensive and compelling set of answers regarding the validity of the multiplicative representation of two reaching direction in the primate motor cortex. After the reviews, the paper is making a large contribution to the knowledge of sequence representation in the voluntary motor system of the monkey.

Thank you once again for all reviewers' comments which have tremendously helped us to improve the manuscript!